# Spatial Analysis of Shared Risk Factors between Pleural and Ovarian Cancer Mortality in Lombardy (Italy)

**DOI:** 10.3390/ijerph19063467

**Published:** 2022-03-15

**Authors:** Giorgia Stoppa, Carolina Mensi, Lucia Fazzo, Giada Minelli, Valerio Manno, Dario Consonni, Annibale Biggeri, Dolores Catelan

**Affiliations:** 1Unit of Biostatistics, Epidemiology and Public Health, DCTVPH, University of Padova, 35131 Padova, Italy; annibale.biggeri@unipd.it (A.B.); dolores.catelan@unipd.it (D.C.); 2Occupational Health Unit, Fondazione IRCCS Ca’ Granda Ospedale Maggiore Policlinico, 20122 Milan, Italy; dario.consonni@unimi.it; 3Department of Environment and Health, Istituto Superiore di Sanità, 00100 Rome, Italy; lucia.fazzo@iss.it; 4Statistical Service, Istituto Superiore di Sanità, 00100 Roma, Italy; giada.minelli@iss.it (G.M.); valerio.manno@iss.it (V.M.)

**Keywords:** asbestos-related diseases, ovarian cancer, pleural cancer, Bayesian shared spatial models, mortality, surveillance, epidemiology

## Abstract

Background: Asbestos exposure is a recognized risk factor for ovarian cancer and malignant mesothelioma. There are reports in the literature of geographical ecological associations between the occurrence of these two diseases. Our aim was to further explore this association by applying advanced Bayesian techniques to a large population (10 million people). Methods: We specified a series of Bayesian hierarchical shared models to the bivariate spatial distribution of ovarian and pleural cancer mortality by municipality in the Lombardy Region (Italy) in 2000–2018. Results: Pleural cancer showed a strongly clustered spatial distribution, while ovarian cancer showed a less structured spatial pattern. The most supported Bayesian models by predictive accuracy (widely applicable or Watanabe–Akaike information criterion, *WAIC*) provided evidence of a shared component between the two diseases. Among five municipalities with significant high standardized mortality ratios of ovarian cancer, three also had high pleural cancer rates. Wide uncertainty was present when addressing the risk of ovarian cancer associated with pleural cancer in areas at low background risk of ovarian cancer. Conclusions: We found evidence of a shared risk factor between ovarian and pleural cancer at the small geographical level. The impact of the shared risk factor can be relevant and can go unnoticed when the prevalence of other risk factors for ovarian cancer is low. Bayesian modelling provides useful information to tailor epidemiological surveillance.

## 1. Introduction

Ovarian cancer is one of the most common cancers among women, ranking seventh in 2018 according to Globocan. It is also important because of its high mortality rate—ovarian cancer is three times more lethal than breast cancer. Worldwide, the incidence was around 300,000 cases and the mortality slightly less than 200,000 deaths in 2018. Approximately 30% of ovarian cancer cases occur in European countries [1]. In Italy, we recorded about 5000 cases and around 3000 deaths per year, and the prevalence was estimated in 50,000 cases in 2020 [2]. 

The most important risk factors in terms of attributable fraction in developed countries are related to reproductive life, obesity, lifestyles, and genetics (among them BRCA mutations). In 2009, the International Agency for Research on Cancer (IARC) concluded that there is sufficient evidence for a causal association between asbestos exposure and ovarian cancer [3]. Apart from the well-known occupational and environmental exposure to asbestos, it is worth mentioning that asbestos was recognized as a contaminant of talc powders. The relative risk of ovarian cancer associated with talc powders was modest (below 1.3) either in case–control [4] or cohort studies [5]. In Italy, consistently with the IARC statement, high standardized mortality ratios (SMR) were reported in a cohort of workers exposed to asbestos, including asbestos-cement plant workers (SMR 1.5) [6] and asbestos textile workers (SMR 3.0) [7]. Ovarian cancer associated with household asbestos exposures is also documented in Italy, particularly in selected areas where environmental pollution from industrial sources was present (SMR 1.42) [8]. These results raised concerns and motivated claims for surveillance programs [9].

Recently, Henley et al. found a positive correlation between malignant mesothelioma (MM) incidence (an indicator of asbestos exposure) and ovarian cancer incidence in the United States [10]. This contribution opens the question of how large the contribution of asbestos exposure to ovarian cancer is at the population level. However, this study did not address the spatial structure of the problem and the uncertainty related to the surrogate use of MM occurrence as a proxy for asbestos exposure.

In previous studies, we analyzed the geographical pattern of incidence of MM by gender in the Lombardy region in 2000–2012 [11]. This study documented high MM occurrence in both genders, attributable to extensive asbestos exposure in the past. In a more recent paper, shared Bayesian models were used to analyze the geographical pattern of MM incidence in men and women in order to disentangle the spatial distribution of MM risk in two components (occupational and environmental) [12]. 

In the present paper, we aimed to evaluate the presence and extent of the risk of ovarian cancer associated with asbestos exposure in a large population (10 million people) by applying a series of Bayesian hierarchical shared models to the bivariate spatial distribution of ovarian and pleural cancer mortality by municipality in the Lombardy Region (Italy) in 2000–2018. Shared Bayesian models were originally developed to study the joint spatial distribution of the risk of different diseases and to evaluate the presence and extent of shared risk factors [13]. In particular, we herein applied Bayesian shared models to try to disentangle environmental and lifestyle determinants of ovarian cancer. This modelling approach is quite general and can be used whenever the inferential goal is the correlation between diseases and shared risk factors.

## 2. Materials and Methods

### 2.1. Mortality and Population Data

We analyzed deaths from ovarian cancer (ICD-9: 183.0, ICD-10: C56) and pleural cancer in females (including pleural cancer, ICD-9: 163 and ICD-10: C38.4 and pleural mesothelioma ICD-10: C45.0) at the municipal level for Lombardy Region (Italy) in the years 2000–2018. Death records were extracted by the Statistical Service of the Italian National Institute of Health (ISS) from the cause-specific mortality database of the Italian National Institute of Statistic (ISTAT). Since ICD-9 codes were still used in Italy during the years 2000–2002, we converted the codes as reported above. Population denominators came from the same sources of mortality data (ISTAT). Death counts and corresponding population denominators for each of the 1546 municipality were categorized into 18 age classes (0–4, …, 85 and more). The annual expected numbers of ovarian and pleural cancer deaths for each municipality were calculated under internal indirect standardization, i.e., by multiplying year- and age-specific reference regional rates to municipality person-years [14]. We then collapsed observed and expected deaths over the whole period to calculate SMRs for each municipality.

### 2.2. Bayesian Models

To study the spatial co-occurrence of the two diseases, we specified a series of Bayesian models. The simple correlation between two diseases at the geographical level is a misleading measure [15] because of the spatially structured random noise always present in geographical or spatial phenomena. Therefore, our analysis strategy was to specify Bayesian models for spatially structured random variables. In this section, we first introduce the popular Besag–York–Mollié model to describe the spatial structure for the two diseases separately, and then we introduce more complex models aimed at assessing the presence and extent of shared risk factors. A section on Bayesian model comparison and model choice completes the methods section.

#### 2.2.1. Spatial Pattern of Diseases 

We describe the spatial pattern of mortality risk for ovarian cancer and pleural cancer in females in the Lombardy Region for 2000–2018 at the municipality level using hierarchical Bayesian models with structured and unstructured random effects (BYM model) [16]. Let us assume the number of observed cases O_ik_ in the i-th municipality (I = 1, …, 1546) and k-th disease (k = “O” Ovarian, “P” Pleural) follow a Poisson distribution with mean E_ik_ × *θ_ik_*, where E_ik_ indicates the expected number of cases under internal indirect standardization [17] and *θ_ik_* is the relative risk taking the regional average as reference. 

A random effects model is specified for the logarithm of the relative risk:(1)$$\log\left(\theta_{ik\;}\right)=\alpha_{k}+u_{ik}+v_{ik}$$
where *α_k_* represents a disease-specific intercept, $u_{ik}$ is a spatially structured (clustering) term by area and disease, and $v_{ik}$ is a spatially unstructured (heterogeneity) term by area and disease. 

Priors for $\alpha_{k}$ are assumed independent and uninformative (improper flat). Gaussian (0, λ_vk_) distributions were specified as prior distributions for the $v_{ik}$ random terms. For the spatially structured random terms $u_{ik}$, conditionally autoregressive priors were adopted. That is, conditionally on u_l~i,k_ (~ i indicates areas adjacent to i-th ones, l = 1, …, 1546), we assumed Gaussian(ū_ik_, λ_uk_ n_i_) distributions, where ū_ik=_ $\sum_{l\sim i}\frac{{\text{ū}}_{\mathrm{lk}}}{n_{i}}$ and n_i_ is the number of the adjacent areas to the i-th one.

The hyperprior distributions for the precision parameters λ_vk_ and λ_uk_ are assumed to be diffuse uninformative Gamma(0.5, 0.0005) [18].

Let us define this model as model 1 (M1): no shared components are present, but only disease-specific clustering and disease-specific heterogeneity random effects are introduced. Let us indicate disease-specific clustering terms as Uk and disease-specific heterogeneity terms as Vk. With this notation, the described model is denoted as M1: Uk Vk.

#### 2.2.2. Shared Bayesian Models

We used several alternative models based on various shared and disease-specific components combinations. In the following, we indicate with U a shared clustering component and with V a shared heterogeneity component. 

Starting from model 1 (M1), we specify eight more models: disease-specific and shared clustering, disease-specific heterogeneity (M2: Uk U Vk); disease-specific, shared clustering, and shared heterogeneity (M3: Uk U V); shared clustering, disease-specific, and shared heterogeneity (M4: U Vk V); disease-specific clustering, disease-specific heterogeneity, and shared heterogeneity (M5: Uk Vk V); shared clustering and shared heterogeneity (M6: U V); disease-specific clustering and shared clustering, disease-specific heterogeneity and shared heterogeneity (M7: Uk U Vk V); shared clustering and disease-specific heterogeneity (M8: U Vk); disease-specific clustering and shared heterogeneity (M9: Uk V).

Let us describe M7, the most complex model (with both specific and shared clustering and heterogeneity terms). Let $O_{iP}$ and $O_{iO}$ denote the observed number of cases in the i-th municipality for pleural and ovarian cancer, respectively. As in the BYM model the likelihoods are $Poisson\left(E_{iP}\theta_{iP}\right)$ and $Poisson\left(E_{iO}\theta_{iO}\right)$, and we specified a log-linear model for the relative risks $\theta_{iP}$ and $\theta_{iO}$:(2)$$\log(\theta_{iP})=\alpha_{p}+u_{i}\times \delta +u_{iP}+v_{i}\times \omega +v_{iP}$$
(3)$$\log\left(\theta_{iO}\right)=\alpha_{O}+u_{i}/\delta +\mu_{iO}+v_{i}/\omega +v_{iO}$$

The terms $v_{i}$ and $u_{i}$ represent the shared heterogeneity and clustering components (i.e., latent components not different by diseases), and $v_{iP},\;\;v_{iO},$ $u_{iP},$ and $u_{iO}$, the unshared ones (i.e., the unique disease-specific random terms). 

The unknown parameters δ and ω are scaling parameters that allow for a different impact of the shared components on each disease. In particular, the effect of a latent shared risk factor—i.e., asbestos exposure, in our example—is allowed to be different on the two diseases. We expect a larger effect on pleural than ovarian cancer, then the tuning parameter should be larger than one—algebraically $\delta =\beta_{P}/\beta_{O}$, where $\beta_{P},\beta_{O}$ are the regression coefficients of the latent shared factor, analogously for ω [19]. They are assumed to be a priori lognormal distributed with zero mean and low precision; in detail, we assigned a N (0, σ^2^) with σ^2^ = 0.17. The priors and hyperpriors for the other model terms are the same specified in the BYM model.

### 2.3. Model Comparison

We evaluated models on two axes: model support (or model fitting) and model robustness (or model influence). Model support is addressed under a Bayesian approach, in terms of predictive accuracy. In brief, we aimed to see how close the predicted number of events are to the observed ones penalized for model complexity. Model robustness is approached by evaluating to which extent the inferences based on a given model differ from those based on the best fitting one. We also aimed to focus on a parsimonious model with easily interpretable model parameters.

We compared the fitting of the different models using the *WAIC* criterion [20]. *WAIC* is a measure of predictive accuracy for the set of data points taken one at a time, approximating the expected log pointwise predictive density.

In detail,
(4)$$\mathrm{WAIC}=-2lppd+2p_{\mathrm{WAIC}}$$
where
(5)$$lppd=\underset{i}{\sum }\log\left\{\int p\left(y_{i}|\theta \right)p\left(\theta |y\right)d\theta \right\}$$
is the log pointwise predictive density and
(6)$$p_{WAIC}=\sum_{i}VAR\left[\log\left(p\left(y_{i}\right)|\vartheta \right)\right]$$
is the variance of log predictive density.

Using posterior draws from the MCMC chains, we computed *WAIC* as
(7)$$WAIC=-2\overset{n}{\underset{i=1}{\sum }}\log\left(\frac{1}{K}\overset{K}{\underset{k=1}{\sum }}p\left(y_{i}|\theta^{k}\right)\right)+2\frac{1}{K-1}\overset{K}{\underset{k=1}{\sum }}{\left[\log p\left(y_{i}|\theta^{k}\right)-\frac{1}{K}\overset{K}{\underset{k=1}{\sum }}\log\left(p\left(y_{i}|\theta^{k}\right)\right)\right]}^{2}\;$$

We assessed model robustness via a local influence measure. We used the calibrated Kullback–Leibler divergence (*KL*) [21], which assesses how different our inferences would be under plausible alternative models.

This discrepancy can be expressed as the difference in expected utilities between unperturbed and perturbed posteriors (predefining the model for the actual belief). The loss in utility due to the approximation of the predefined model, say *M*_0_, with model *M** is expressed by the Kullback–Leibler divergence (*KL*), where the parameter of interest is the logarithm of the relative risk. We derived a Kullback–Leibler measure separately for each *i*-th observation, then
(8)$$KL_{i}=\int \log\frac{p\left(\theta_{i}\left|Y,M_{0}\right.\right)}{p\left(\theta_{i}\left|Y,M^{*}\right.\right)}p\left(\theta_{i}\left|Y,M_{0}\right.\right)\;d\theta_{i}$$
where $p\left(\theta_{i}\left|Y,M_{0}\right.\right)$ is the posterior marginal distribution of the logarithm of relative risk given data and the model. The posterior distributions of $\theta_{i}$ under *M*_0_ and *M** can be well approximated by normal distributions with mean *m*_0_ and *m** and variances $s_{0}^{2}$ and $s_{*}^{2}$ (the *i* suffix has been dropped for simplicity). The Kullback–Leibler divergence can then be approximated by
(9)$$KL\approx \frac{1}{2}\left[\frac{{\left(m_{*}-m_{0}\right)}^{2}}{s_{*1}^{2}}-1+\frac{s_{0}^{2}}{s_{*1}^{2}}-\log\left(\frac{s_{0}^{2}}{s_{*}^{2}}\right)\right]$$

McCulloch (1989) [21] suggested calibrating Kullback–Leibler divergences setting *KL* = *c* and finding that value *p*(*c*) such that
(10)$$KL\left[B\left(0.5\right),B\left(p\left(c\right)\right)\right]=c$$
where *B*(*p*) is the Bernoulli distribution with parameter *p*, and the value *p*(*c*) is the calibrated Kullback–Leibler divergence. Since we had one calibrated *KL* for each observation, we summarized the information of model influence by the median calibrated *KL*.

In this work, we traded off models using *WAIC* vs. median calibrated *KL*, taking as reference the most supported model, in our case, model 7 (M7) [22,23,24]. 

### 2.4. Computational Details

For all the models described, the marginal posterior distributions of the parameters of interest were approximated by Monte Carlo Markov chain (MCMC) methods. We used WinBUGS software [25] to perform the MCMC analyses. For each model, we ran two independent chains; checks for achieved convergence of the algorithm were performed following Gelman and Rubin (1992) [26].

## 3. Results

The number of deaths for pleural cancer among women in the period 2000–2018 in Lombardy Region (Italy) was 2070, with the following distribution of ICD codes: ICD9: 163 (malignant neoplasm of the pleura)—262 deaths (2000–2002); ICD10: C38.4 (malignant neoplasm of the pleura)—271 deaths; ICD10: C45.0 (mesothelioma of the pleura)—1537 deaths. Across municipalities, SMRs ranged from 0 to 22.14. The expected number of deaths was highly variable, ranging from less than 0.01 to 333.20, reflecting the large variability in population size (maximum: Milan, the regional capital, 1,351,562 inhabitants; minimum: Morterone, 29 inhabitants). The municipality of Broni, in the province of Pavia, showed the second highest SMR equal to 18.3 (observed 51 cases; expected: 2.79). The highest SMR was observed in a very small municipality with 0.05 expected cases.

Regarding ovarian cancer, we retrieved 10,462 deaths, with SMRs ranging from 0 to 19.15. Most of the high SMRs were very unstable, being based on less than five expected cases. 

Figure 1A shows the histogram of SMRs of the two diseases, and Figure 1B shows the funnel plot of SMRs vs. expected number of cases with one-sided confidence bands [27]. 

The five municipalities with one-sided *p*-values < 0.001 and SMR > 1 for ovarian cancer are listed in Table 1. Notably, three of them also had high SMR of pleural cancer. 

### 3.1. Spatial Pattern of the Two Diseases

In Figure 2, we report the spatial distribution of SMRs (upper panels) and smoothed RRs (lower panels) from the BYM model for pleural cancer (left panels) and ovarian cancer (right panels). The shrinkage effect of Bayesian RRs from the BYM model was evident compared with SMRs. The geographical pattern for pleural cancer was very strong. In addition, well-known asbestos-polluted industrial areas were clearly identified [11,12].

The spatial pattern for ovarian cancer was more smoothed, although some overlapping areas with high pleural cancer risks can be identified, as shown in the smoothed RR map on a relative scale (Appendix A). 

Table 2 reports the mean of the posterior distribution of the heterogeneity and clustering hyperparameters (expressed as standard deviations (SD) of *u**ik* and *v**ik*).

The heterogeneity/clustering odds illustrates that pleural cancer spatial pattern was, as expected, strongly spatially structured; conversely, ovarian cancer showed a more complex pattern with both random components almost equally represented. Note that there were examples of diseases with only heterogeneity component and almost no clustering component; instead, interestingly, ovarian cancer showed a non-negligible clustering term. [28]. This observation motivated the following analysis, in which we explored how much the pleural cancer clustering component was shared with the ovarian cancer clustering component.

### 3.2. Shared Bayesian Models and Model Comparison

The BYM model is based on a prior belief that the two diseases do not share common risk factors and show their own specific spatial pattern. Therefore, we labelled this model as model 1 (M1). 

We considered as an alternative a model with both specific and shared random components, model 7 (M7).

We can specify several other models between these two, dropping shared or specific heterogeneity or clustering components. Each model had its own interpretation. To illustrate this on models M1 and M7, we report in Figure 3 the log-log scatter plot of SMRs and Bayesian RRs for ovarian and pleural cancer, for M1 and M7. Under M1 (panel (A)), no association—even a negative one—was evident between ovarian and pleural cancers. Under M7 (panel (B)), a positive association was evident, as highlighted by the regression lines of the Bayesian RRs of ovarian cancer on the Bayesian RRs of pleural cancer. 

However, as shown in Figure 3, SMRs were extremely dispersed. Therefore, it is not easy to find a pattern or to judge model adequacy on a visual inspection of the raw data, and we must rely on more complex model comparison. 

Table 3 reports the details of each model, assuming different association structures between ovarian and pleural cancer, and the estimates of *WAIC* and median calibrated *KL*, respectively. Figure 4 reports the graph of *WAIC* vs. median calibrated *KL*. Remember that M7 (the more complex and best supported model) is taken as reference.

There is no best model in a Bayesian perspective: alternative perspectives must be explored [29]. The most supported model was, as expected, M7. However, M4 appeared to be a good parsimonious alternative with the advantage of better interpretability of the shared spatial pattern between pleural and ovarian cancer mortality.

Under model 4, the shared clustering component U (Figure 5 left panel) highlighted areas with a common risk factor for the two diseases. These areas are well-known areas where asbestos exposure was present [12]. 

In Figure 6, we report the joint posterior distributions of the shared clustering component $\frac{1}{\delta }\times U$ and the specific heterogeneity term *V_o_* for three selected municipalities that were very different in terms of risk for pleural and ovarian cancer. To explain the figure, let us start recalling the linear predictor of ovarian cancer of the more complex model M7:(11)$$\log\left(\theta_{iO}\right)=\alpha_{O}+u\;/\;\delta +u_{iO}+v_{i}\;/\;\omega +v_{iO}\;$$

Now, model 4 is simpler because it dropped the clustering specific term: $u_{iO}\;\left\{⬚\right\}$. To show the relative contribution of shared and specific components to the relative risk, we report in Figure 6 the specific heterogeneity term of ovarian cancer $v_{iO}$—related to specific risk factors—and the shared components $u_{i}\;/\;\delta +v_{i}\;/\;\omega$—related to shared risk factors with pleural cancer. Moreover, note that the posterior median and credibility intervals of the tuning parameter δ were 0.265; 95%CrI 0.197–0.342, and for ω were 0.659; 95%CrI 0.432–1.036. This means that the relative risk of pleural cancer for asbestos exposure was four times (or 1.5 times) the relative risk of ovarian cancer for the shared clustering component (shared heterogeneity component). Let us consider the green points in Figure 6, which refer to the municipality of Broni, which had a high risk for pleural cancer but not for ovarian cancer. Consistently, the shared clustering component had, on average, positive values. In contrast, on average, the specific heterogeneity term was negative (the values were on the log RR scale, and thus positive values of means RR above the null and negative values RR below the regional mean). The municipality of Calcio (in orange) was characterized by positive values of both components, consistently with a higher relative risk for pleural and ovarian cancers. The shared clustering component of the municipality of Rosate (in blue) was centered around values below zero—consistent with the lower risk of pleural cancer. 

Lastly, we checked for residuals by municipality. As expected, Bayesian RRs were shrunk toward the null value, and therefore we found large positive residuals for relatively small municipalities with higher SMRs (Figure 7, points with Pearson’s residuals greater than 2). The municipality of Broni, at the highest risk for pleural cancer, was not an outlier for ovarian cancer, even if the Bayesian RR was 1.23 and the SMR was 0.97—observed cases 13 and 16.5 predicted cases. When fitting models with spatially structured random terms, the posterior estimates were also shrunk toward the local mean, and it was not unexpected that the posterior RRs went in the opposite direction of SMRs.

## 4. Discussion

Studying the spatial distribution of asbestos-related diseases (ARD) is important to define surveillance programs at the regional or national level [30]. Ovarian cancer is considered one of the diseases related to asbestos exposure since the IARC assessment [3], and it is a matter of debate if it has to be included in epidemiological surveillance of asbestos exposed populations.

Given the high etiological fraction attributable to asbestos exposure, mortality from malignant mesothelioma and asbestosis is commonly used to estimate the health impact of asbestos at the national and global levels [30,31]. Accordingly, the geographical distribution of malignant mesothelioma (MM) can be used as a proxy of past asbestos exposure [32,33,34].

Some recent papers in the literature investigated the spatial co-occurrence of ovarian cancer and MM [10]. There is some dispute about potential misclassification between peritoneal mesotheliomas and ovarian cancer [35]. We think that this problem is a minor one in the case of ovarian cancer. In particular, since ovarian cancer is about 100 times more frequent than peritoneal mesothelioma (in the present study, 10,480 and 100 cases, respectively), a misdiagnosis of even a small fraction of ovarian cancers could spuriously increase the frequency of peritoneal cancers. Conversely, even a misclassification of a large fraction of peritoneal mesothelioma would have a negligible impact on ovarian cancer rate. In any case, we preferred to avoid this problem by analyzing pleural cancer and not all mesotheliomas. The Appendix A report the analysis on MM (all sites, 2439 deaths) and ovarian cancer. Results were quite similar. The potential misclassification of ovarian cancer as peritoneal MM might explain to the slightly higher correlation between the two diseases.

However, mortality from all pleural cancers can still be subject to misclassification, and, when the purpose is the estimation of the etiologic fraction, the number of cases attributable to asbestos exposure may be overestimated [36].

The Bayesian models we specified can be considered bivariate (two-disease) models, the main objective being to evaluate the presence of shared factors in the occurrence of the two diseases. We interpreted the shared factors as proxies of the prevalence of asbestos exposure. This approach can also be viewed as a regression model with an explanatory variable measured with error (in our case, pleural cancer as a proxy of asbestos exposure). For a detailed discussion of the interpretation of shared components, see Knorr-Held Best (2000) [19].

The correlation between two risk maps is difficult to assess, and the use of the correlation coefficient can be misleading [15]. Spatially structured underlying factors may produce spurious correlations. Moreover, the ecological fallacy is inherent in any analysis of aggregate data. Our approach made a step forward because we based the analysis on small geographical units, and we specified spatially structured and unstructured random terms to adjust for hidden confounders. Conditional on these disease-specific random terms, we specified shared random terms, which are interpretable as shared risk factors between the two diseases. Several models can be specified, and model comparison is important in our paper. According to the recent Bayesian literature, model adequacy is evaluated using a measure of predictive accuracy. The idea is to evaluate how well a model predicts future data, and we used the *WAIC* statistics for this purpose—penalizing for model complexity. *WAIC* has been widely used in disease mapping [37,38,39]. 

In our application, the empirical data are noisy and difficult to summarize: as exemplified in Figure 3, two different models provided contrasting results on the correlation between the two diseases. The observed data did not help prima vista. Filtering random noise and isolating systematic patterns is not trivial, and it is not easy to decide which model should be preferred and how to report uncertainty. We chose to address model uncertainty evaluating how different would be our inferences under plausible alternative models. We used the Kullback–Leibler (*KL*) divergence between each alternative model and the best fitting one (the more complex model 7). If our results were not robust, then we would have found some alternative models with close goodness of fit (*WAIC*) but large *KL* divergence, representing almost equally supported alternative views. As shown in Figure 3, model M1 yielded negative associations between ovarian and pleural cancer, while positive associations were found under model M7. There was a large discrepancy between results from these two models, as measured by the *KL* divergence. If in case of similar goodness of fit, our inference would not be robust and the course of action would be clear but dull—i.e., it would be necessary to report the two different points of view. Luckily, model M1 showed substantially worse support than model M7. As shown in Figure 4, there was evidence of model robustness, and some plausible alternative models with similar *WAIC* to model 7 provided the same picture with small *KL* divergence. Indeed, to simplify the interpretation of the results, we focused on the more parsimonious model 4 [22]. The interpretation here was simpler because model 4 included only one clustering term, the shared component between the two diseases. These shared components can be easily interpreted as the underlying prevalence of asbestos exposure corresponding to the spatial pattern of pleural cancer (see Figure 5). 

A further justification for a parsimonious model is that it is difficult to disentangle the contribution of each random term in a Bayesian spatial random effect model. For example, regarding the M1 BYM model, it is not recommended to report the contribution of the clustering and heterogeneity components separately [18]. In general, there is an identifiability issue and Bayesian models are overparametrized [40]. As an example of such difficulties, we can show for the municipality of Milan, the capital city, that there was a strong collinearity between the specific and shared heterogeneity terms in model 4, and that therefore it is impossible to separate the contribution of these two terms. The shared clustering term for Milan was consistently estimated with value close to zero—the SMR for ovarian cancer and pleural cancer were, respectively, 1.01 and 0.96, as shown in Figure 8. The shared clustering model M4 provide stable estimates of the shared clustering terms due to the information from the two diseases (see Figure 5).

However, beyond identifiability, there is also model uncertainty to be discussed. A map of calibrated *KL* by municipalities for model 4 (Appendix A) showed lower robustness in the Broni area. The behavior of the different models was clearly shown when looking at the three different municipalities of Calcio, Rosate, and Broni, two with strong documented asbestos exposure (Calcio and Broni) and two with high background risk of ovarian cancer (Calcio and Rosate).

For the municipalities of Calcio and Rosate, the shared and specific terms were clearly identified and showed the different contributions of risk factors, without inconsistencies among different models (Figure 6).

For the municipality of Broni, the SMR for pleural cancer was the second highest, while for ovarian cancer was 0.97, slightly below the regional average (13 observed cases, 13.37 expected). Model 4 predicted a risk of ovarian cancer above the regional average (RR 1.23, 95% CrI 1.01; 1.49). The predicted observed cases by model M4 were 16.52 with a Pearson’s residual that was not extreme (i.e., between −2 and 2) (Figure 7). Conversely, the estimates were 0.93 (95% CrI 0.80; 1.06) under the less supported model 1 and 1.06 (95% CrI 0.80; 1.37) under the best supported model 7. Credibility intervals of model M4 and model M7 were almost two times wider than of model M1. For pleural cancer in Broni, we did not find any difference among the different models because of pleural cancer’s strong spatially structured pattern. In this case, the reader should be cautious in making an inference on the risk of ovarian cancer in Broni, and look carefully at sensitivity analysis.

In summary, analyses suggest that the ovarian cancer risk in Broni is a combination of low prevalence of risk factors for ovarian cancer and high prevalence of asbestos exposure. 

## 5. Conclusions

We found evidence of a shared risk factor between ovarian and pleural cancer mortality at small geographical level. The impact of the shared risk factors on of ovarian cancer rates at the population level can be relevant. In areas where the prevalence of other risk factors for ovarian cancer is low, the risk of ovarian cancer tends to be low, even if there is a substantial asbestos exposure. The reason is that asbestos attributable fraction for ovarian cancer is low and population prevalence of asbestos exposure among women is not high.

This is true on average, but it can be very different in specific areas where the prevalence of asbestos exposure was very high. Therefore, the burden of ovarian cancer due to asbestos exposure can go unnoticed in small areas at low background risk of ovarian cancer. The well-documented high MM rate in Broni (among the highest in Italy) provides an example of such difficulties. Bayesian modelling provides useful information to tailor epidemiological surveillance. Bayesian modelling is sensitive, allows incorporation of previous knowledge, and provides adequate treatment of uncertainty and model robustness.

## Figures and Tables

**Figure 1 ijerph-19-03467-f001:**
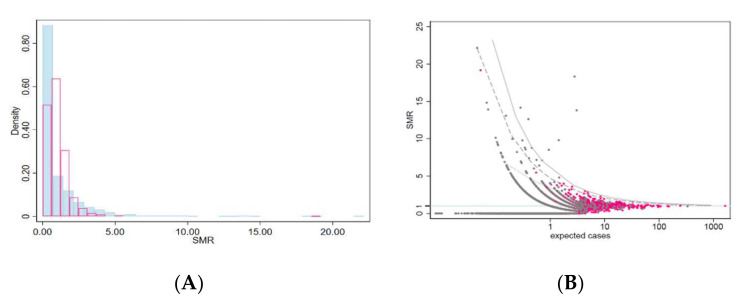
(**A**) Histogram of standardized mortality ratios (SMR) by municipality (pleural cancer (light-blue) and ovarian cancer (pink)). Lombardy Region, 2000–2018. (**B**) Funnel plot of SMR for pleural cancer (grey) and ovarian cancer (pink) by municipality vs. expected number of cases (log scale). One-sided confidence bands 0.99 (dotted line); 0.999 (dashed line); 0.9999 (solid line).

**Figure 2 ijerph-19-03467-f002:**
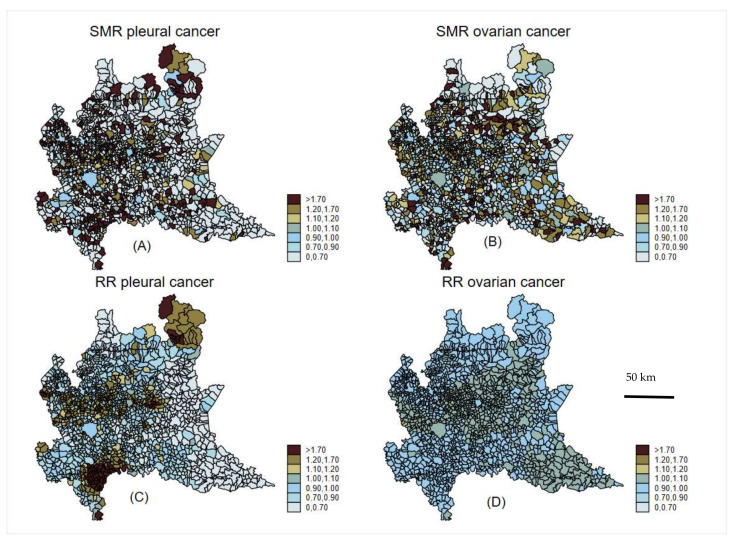
Spatial choropleth map of raw (SMR pleural cancer (**A**); RR ovarian cancer (**B**)) and Bayesian smoothed (RR pleural cancer (**C**); RR ovarian cancer (**D**)) standardized mortality ratios of pleural cancer and ovarian cancer. Lombardy Region, 2000–2018. Absolute scale.

**Figure 3 ijerph-19-03467-f003:**
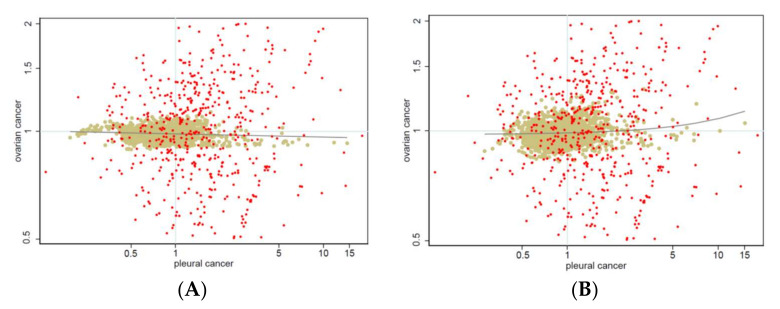
Log–log scatter plot of raw (red dots) and Bayesian smoothed standardized mortality ratios (SMR) (khaki dots) for ovarian cancer (y-axis) and pleural cancer (x-axis). (**A**) Estimates from M1; (**B**) estimates from M7 (see text). Linear regression lines fitted on Bayesian smoothed rates. Lombardy Region, 2000–2018. Absolute scale.

**Figure 4 ijerph-19-03467-f004:**
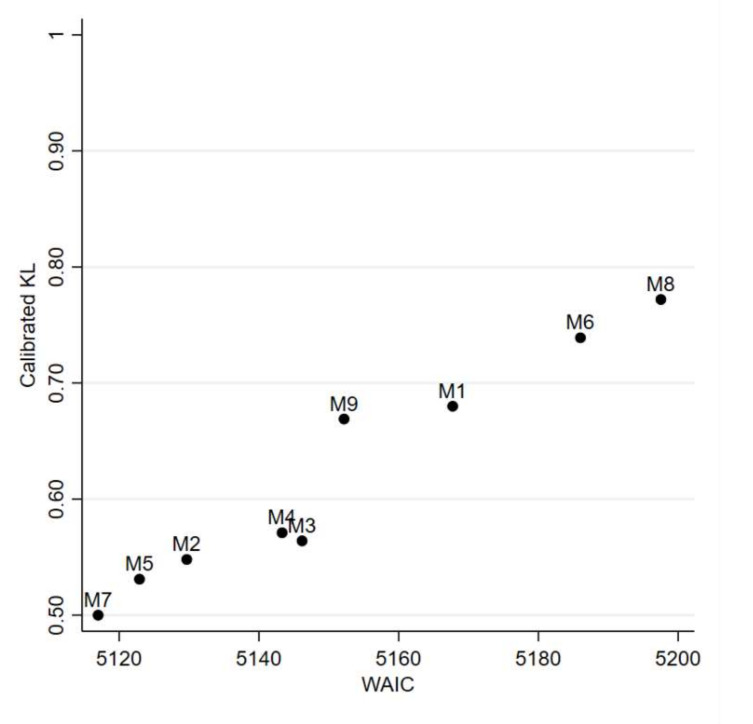
Scatter plot of median calibrated Kullback–Leibler divergence (calibrated *KL*) vs. predictive accuracy measured by the widely applicable or Watanabe–Akaike information criterion (*WAIC*) for each fitted Bayesian model. Calibrated *KL* was calculated with regard to model 7—the more complex and best-supported model. Ovarian cancer and pleural cancer mortality by municipality. Lombardy Region, 2000–2018.

**Figure 5 ijerph-19-03467-f005:**
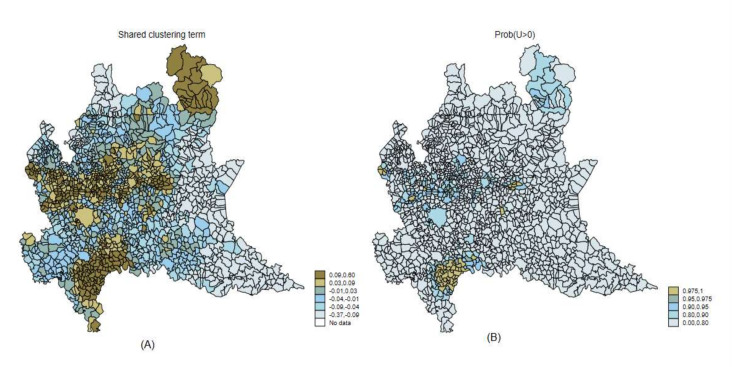
Maps of shared clustering terms (U) between pleural cancer and ovarian cancer from model 4 (M4) (**A**) and Posterior Probability of Direction of Effect—Prob (U > 0) (**B**). Lombardy Region, 2000–2018.

**Figure 6 ijerph-19-03467-f006:**
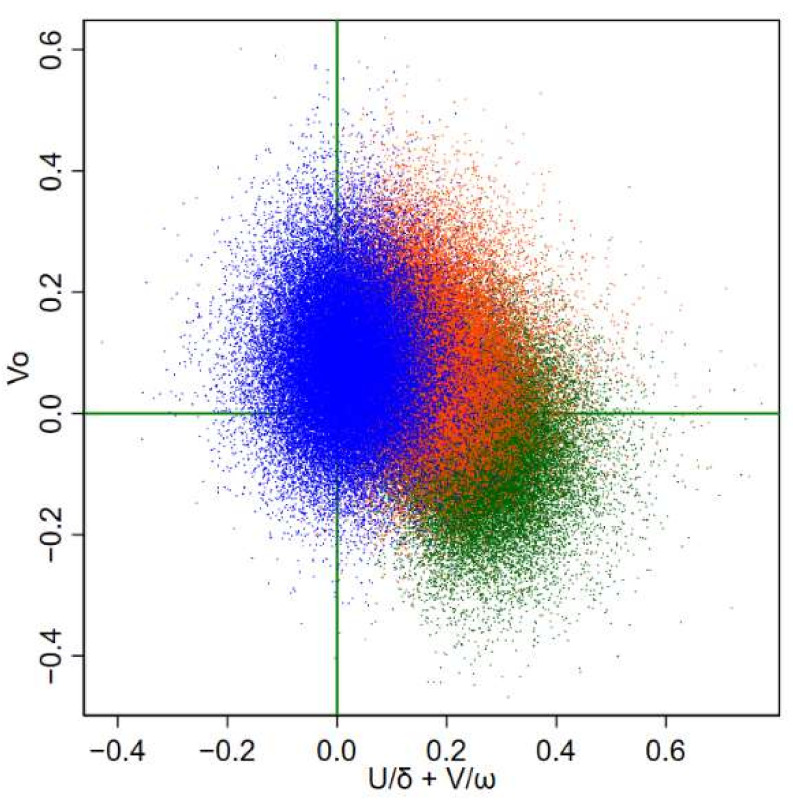
Scatter plot of the specific heterogeneity terms vs. shared components ($\frac{1}{\delta }\times U+\frac{1}{\omega }\times V$) of ovarian cancer from model 4 (M4) in four municipalities. Lombardy Region, 2000–2018. Blue: Rosate; orange: Calcio; green: Broni. Fifty thousand draws from the joint posterior distribution.

**Figure 7 ijerph-19-03467-f007:**
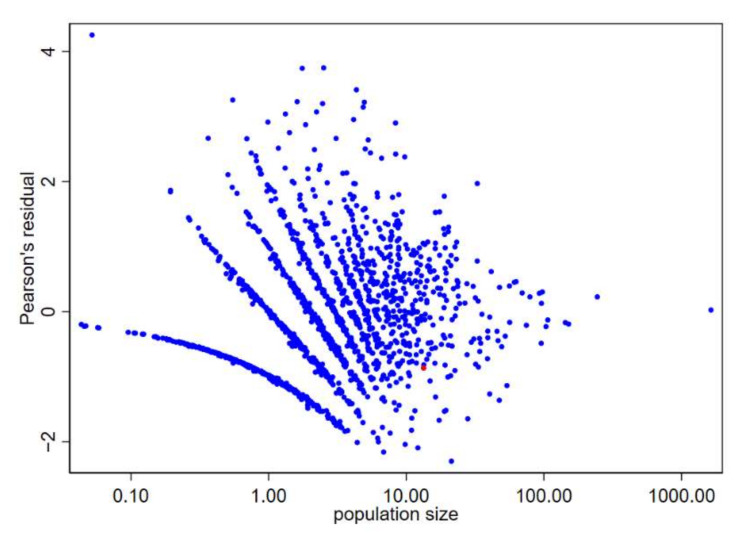
Scatter plot of Pearson’s residuals vs. population size (expressed as expected number of cases based on internal indirect standardization) by the municipality. Ovarian cancer. Lombardy Region, 2000–2018. Broni municipality shown in red.

**Figure 8 ijerph-19-03467-f008:**
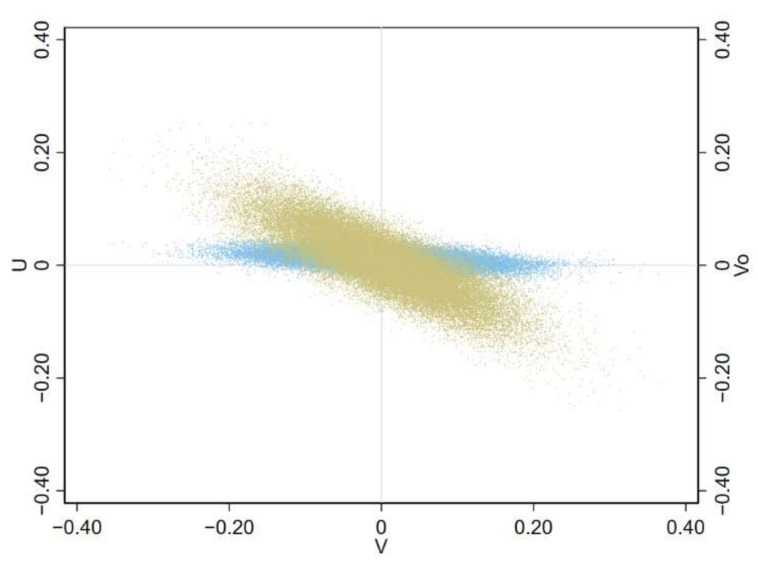
Scatter plot of the shared clustering and specific heterogeneity terms vs. shared heterogeneity term of ovarian cancer from model 4 (M4) for the municipality of Milan. Lombardy Region, 2000–2018 (light blue: shared clustering; khaki: specific heterogeneity). Fifty thousand draws from the joint posterior distribution.

**Table 1 ijerph-19-03467-t001:** Observed and expected deaths, and standardized mortality ratios (SMR) for ovarian cancer and pleural cancer. Municipalities with SMR for ovarian cancer above the regional average (*p* < 0.001). Lombardy Region, 2000–2018.

Municipality	Ovarian Cancer			Pleural Cancer		
	Observed	Expected	SMR	Observed	Expected	SMR
	Deaths	Deaths		Deaths	Deaths	
Rosate	13	4.96	2.62	0	0.94	0
Bovegno	9	2.51	3.58	<3	0.50	3.97
Capriate SG	19	8.37	2.27	<3	1.66	1.2
Pedrengo	13	4.35	2.99	3	0.79	3.81
Calcio	14	4.87	2.87	8	0.94	8.52

**Table 2 ijerph-19-03467-t002:** Mean of the posterior distribution of the heterogeneity and clustering hyperparameters and the heterogeneity/clustering ratio.

Disease	Clustering SD	Heterogeneity SD	Odds Heterogeneity: Clustering
Pleural cancer	0.6267	0.1207	1:5.2
Ovarian cancer	0.08213	0.0567	1:1.4

**Table 3 ijerph-19-03467-t003:** *WAIC* and median calibrated Kullback–Leibler divergence (Calibrated *KL*) for all the fitted models.

Model	Description	*WAIC*	Calibrated *KL*
M7	Uk U Vk V	5116.985	0.500
M5	Uk Vk V	5122.904	0.531
M2	Uk U Vk	5129.669	0.548
M3	Uk U V	5146.176	0.564
M4	U Vk V	5143.311	0.571
M9	Uk V	5152.181	0.669
M1	Uk Vk	5167.727	0.680
M6	U V	5186.008	0.739
M8	U Vk	5197.524	0.772

## Data Availability

Data were provided by the Italian National Institute of Statistic (ISTAT) and cannot be made available by the authors.

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
