# Peer review of "Spatial Analysis of Shared Risk Factors between Pleural and Ovarian Cancer Mortality in Lombardy (Italy)"

_ijerph, 2022, doi:10.3390/ijerph19063467_

Round 1

Reviewer 1 Report

This is a well-conducted and well-described analysis of a challenging problem in environmental health. The methods are complex and innovative, and the findings provide good evidence that this approach could find many applications in other settings and with other problems of correlated disease patterns.

There are some minor concerns that should be addressed to improve comprehension. There are many small errors of English that can be corrected by close editing by a native speaker. I have not noted all of these here.

Minor concerns

Line 72-73. I think it would help motivate readers to work through the methods if you could provide another sentence here about what problem the shared Bayesian models solves.

Line 141. The most complex model….

Line 157. I do not understand the nomenclature here: sigma squared equals 0:17?

Line 248. Figure 2. I think it would be helpful if you would provide descriptive labels to the 4 maps, so that the reader doesn’t have to refer so closely to the figure legend when examining them. Similar point for Figure 5.

Lines 369 – 371. I don’t understand this sentence.

Lines 377-378. “doesn’t solve all the problems”. This is not clear to me.

Line 414. “Dull but clear” is not a meaningful expression in English.

Line 421. “Models of pleural cancer include mainly clustering terms.” I don’t understand this.

Lines 424-425. Please explain why “it is not recommended…”

Line 454. Please explain what we are meant to understand by this sentence – what is the significance here of a Pearson residual that is not extreme?

Author Response

Reviewer 1

This is a well-conducted and well-described analysis of a challenging problem in environmental health. The methods are complex and innovative, and the findings provide good evidence that this approach could find many applications in other settings and with other problems of correlated disease patterns.

There are some minor concerns that should be addressed to improve comprehension. There are many small errors of English that can be corrected by close editing by a native speaker. I have not noted all of these here.

Response: We thank the referee for the careful and insightful review of our manuscript. Below we answer to the raised concerns:

Minor concerns

1- Line 72-73. I think it would help motivate readers to work through the methods if you could provide another sentence here about what problem the shared Bayesian models solves.

1- Reply: We added the following sentences : “In particular, here we applied Bayesian shared models to try to disentangle environmental and lifestyle determinants of ovarian cancer. This modelling approach is quite general and can be used whenever the inferential goal is the correlation between diseases and shared risk factors.”(lines 75-79)

2- Line 141. The most complex model….

2- Reply: We amended the text as suggested.

3- Line 157. I do not understand the nomenclature here: sigma squared equals 0:17?

3- Reply: We apologize, it was a typo.We corrected the text (σ2 = 0.17).

4- Line 248. Figure 2. I think it would be helpful if you would provide descriptive labels to the 4 maps, so that the reader doesn’t have to refer so closely to the figure legend when examining them. Similar point for Figure 5.

4- Reply: Thanks for this suggestion. We added descriptive labels to Figures 2 and 5.

5- Lines 369 – 371. I don’t understand this sentence.

5- Reply: We agree with the reviewer and we revised the sentence as follows: “There is some dispute about the impact of potential misclassification between peritoneal mesotheliomas and ovarian cancer [35]. We think that this problem is a minor one in case of ovarian cancer. In particular, since ovarian cancer is about 100 times more frequent than peritoneal mesothelioma (in the present study 10,480 and 100 cases, respectively), a misdiagnosis of even a small fraction of ovarian cancers could spuriously increase the frequency of peritoneal cancers. Conversely, even a misclassification of a large fraction of peritoneal mesothelioma would have a negligible impact on ovarian cancer rate. In any case, in the present paper we preferred to avoid this problem by analyzing pleural cancer and not all mesotheliomas.”(lines 399-408)

6- Lines 377-378. “doesn’t solve all the problems”. This is not clear to me.

6- Reply: We have revised the sentence as follows: “However, mortality from all pleural cancers can still be subject to misclassification and, when the purpose is the estimation of the etiologic fraction, the number of cases attributable to asbestos exposure may be overestimated [36].” (lines 416-418)

7- Line 414. “Dull but clear” is not a meaningful expression in English.

7- Reply: We apologize for the confusion. The citation is “clear but dull” from Simpson’s paper (1951) and it is referred to the interaction term in the analysis of a contingency table. In our paragraph if two models had the same data support we should report model output for both of them, in the same way of when we have a significant interaction term and we should report stratum specific estimates.

8- Line 421. “Models of pleural cancer include mainly clustering terms.” I don’t understand this.

8- Reply: We rephrase the sentence as follows:The interpretation here is simpler because model 4 includes only one clustering term, the shared component between the two diseases. This shared component is easily interpreted as the underlying prevalence of asbestos exposure corresponding to the spatial pattern of pleural cancer. (see Figure 5).” (lines 464-468)

9- Lines 424-425. Please explain why “it is not recommended…”

9- Reply: It is not recommended because they are not completely identifiable (Eberly, L.E. and Carlin, B.P. (2000) Identifiability and Convergence Issues for Markov Chain Monte Carlo Fitting of Spatial Models. Statistics in Medicine, 19, 2279-2294.(citation included in the revised paper) (line 451)

10- Line 454. Please explain what we are meant to understand by this sentence – what is the significance here of a Pearson residual that is not extreme?

10- Reply: Usually outlying residuals have values above 2 or below -2. The red point (Broni municipality) corresponds to a Pearson’s residual value which is not outside these limits.

Reviewer 2 Report

The authors attempted to quantify and map the associations between asbestos exposure on one side and ovarian and pleural cancers on the other in Lombardy Region between 2000-2018 using a spatially explicit Bayesian modelling approach. Areal disease mapping using a Bayesian statistical framework is indeed one of the most robust approaches for guiding disease surveillance efforts. The manuscript has the novelty and relevance to the field, and statistical methods were appropriate and robust. Indeed, tables and figures were presented excellently, and the discussion was rigorous and very deep in interpreting and explaining the results. Very informative paper, and I think it is suitable for publication in IJERPH, after further linguistic improvements.

Author Response

Reply: Thank you for the positive consideration reserved to our work. The manuscript has been edited by an English-speaking native, so we hope it now matches the journal standard.

Reviewer 3 Report

1. Check the style of equations in line with the standard of the journal.
2. Figure 2 and 5 could be plotted in colors and with a scale bar. Multiple panels should be listed as: (a), (b), (c), and so on. 

Author Response

Reviewer 3

  1. Check the style of equations in line with the standard of the journal.

Reply: We made changes following the standard of the journal. All equations were edited using Microsoft Equation Editor.

  1. Figure 2 and 5 could be plotted in colors and with a scale bar. Multiple panels should be listed as: (a), (b), (c), and so on.

Reply: We have amended the figures following the reviewer comments. We added scale bars.

Reviewer 4 Report

The article "Spatial analysis of shared risk factors between pleural and ovarian cancer mortality in Lombardy (Italy)" analyzes the spatial distribution of pleural and ovarian cancer mortality in Lombardy, 2000-2018, using Bayesian hierarchical models. Clustering of pleural cancer deaths was much clearer than ovarian cancer deaths. Even so, the authors were able to demonstrate a possible shared risk factor at small geographical units. 

Comments:

  1. Please provide numbers for the distribution of the 2,070 pleural cancer deaths according to the registered ICD codes 163, 38.4 and C45.0
  2. The findings of ovarian cancer deaths in the Broni are puzzling. Are there any data for the prevalence of other risk factors in the municipality that can be added in the discussion? Is it possible that Broni, as a "hot spot" for asbestos-related diseases, may have overdiagnosis of peritoneal MM?

Author Response

Reviewer 4

The article "Spatial analysis of shared risk factors between pleural and ovarian cancer mortality in Lombardy (Italy)" analyzes the spatial distribution of pleural and ovarian cancer mortality in Lombardy, 2000-2018, using Bayesian hierarchical models. Clustering of pleural cancer deaths was much clearer than ovarian cancer deaths. Even so, the authors were able to demonstrate a possible shared risk factor at small geographical units.

Comments:

Please provide numbers for the distribution of the 2,070 pleural cancer deaths according to the registered ICD codes 163, 38.4 and C45.0

Reply: The distribution of the 2070 pleural cancer deaths is reported below and in the text:

ICD9 163 (Malignant neoplasm of the pleura):      262 deaths (2000-2002)

ICD10 C38.4 (Malignant neoplasm of the pleura): 271 deaths

ICD10 C45.0 (Mesothelioma of the pleura): 1537 deaths

The findings of ovarian cancer deaths in the Broni are puzzling. Are there any data for the prevalence of other risk factors in the municipality that can be added in the discussion? Is it possible that Broni, as a "hot spot" for asbestos-related diseases, may have overdiagnosis of peritoneal MM?

Reply: We agree with the referee that the findings are somewhat puzzling. However, we had discussed them as follows.

1) In Lines 355-361 we commented: “Let’s consider the green point in the Figure 6 (the municipality of Broni), which had a high risk for pleural cancer but not for ovarian cancer. Consistently, the shared clustering component has, on average positive values. In contrast, on average the specific heterogeneity term is negative (the values are on the log RR scale, so positive values means RR above the null and negative values RR below the regional mean).” The estimated RR of ovarian cancer for Broni is 1.24, an estimate which appears internally consistent (see Lines 375-377 and Figure 7).

2) Moreover, we spent a paragraph in the discussion session on the uncertainty regarding the estimate of RR of ovarian cancer for Broni. (Lines 498-511)

3) Finally, we concluded: “Our findings are that analyses suggest that the ovarian cancer risk in Broni is a combination of the low prevalence of risk factors for ovarian cancer and high prevalence of asbestos exposure”. (Lines 512-514)

Regarding potential misclassification we have added the following sentence in the discussion:

“In particular, since ovarian cancer is about 100 times more frequent than peritoneal mesothelioma (in the present study 10,480 and 100 cases, respectively), a misdiagnosis of even a small fraction of ovarian cancers could spuriously increase the frequency of peritoneal cancers. Conversely, even a misclassification of a large fraction of peritoneal mesothelioma would have a negligible impact on ovarian cancer rate.” (lines 401-406)